# Collective dipole effects in ionic transport under electric fields

N. Salles [1,2✉], L. Martin-Samos [1✉], S. de Gironcoli[3], L. Giacomazzi[1,2], M. Valant [2], A. Hemeryck [4], P. Blaise[5], B. Sklenard [5] & N. Richard [6]

In the context of ionic transport in solids, the variation of a migration barrier height under electric fields is traditionally assumed to be equal to the classical electric work of a point charge that carries the transport charge. However, how reliable is this phenomenological model and how does it fare with respect to Modern Theory of Polarization? In this work, we show that such a classical picture does not hold in general as collective dipole effects may be critical. Such effects are unraveled by an appropriate polarization decomposition and by an expression that we derive, which defines the equivalent polarization-work charge. The equivalent polarization-work charge is not equal neither to the transported charge, nor to the Born effective charge of the migrating atom alone, but it is defined by the total polarization change at the transition state. Our findings are illustrated by oxygen charged defects in MgO and in $SiO_2$.

[1] CNR-IOM/Democritos National Simulation Center, Istituto Officina dei Materiali, c/o SISSA, via Bonomea 265, Trieste 34136, Italy. [2] Materials Research Laboratory, University of Nova Gorica, Vipavska 11c, 5270 Ajdovščina, Slovenia. [3] SISSA, via Bonomea 265, Trieste 34136, Italy. [4] LAAS-CNRS, Université de Toulouse, CNRS, Toulouse, France. [5] Univ. Grenoble Alpes, CEA, LETI, Grenoble 38000, France. [6] CEA, DAM, Arpajon 91297, France. ✉email: nsalles33@gmail.com; marsamos@iom.cnr.it

on motion in solids proceeds via jumps between different sites. In the absence of an external electric field, the probability that an ion will jump is proportional to the Boltzmann weight of the migration barrier. The field will, however, lower the barrier for motion along its direction, thus increasing the probability of movement and, therefore, the drift velocity of the ion.

Very recently, it has been formally demonstrated that the transported charge—the charge that is carried from one site to a symmetrically equivalent site (displacement of an atomic sublattice by a lattice vector)—is an integer number, thus providing a sound definition of the oxidation state of a migrating ion (see ref. [1]). It might therefore appear natural to assume that the charge involved in the barrier height variation is the same charge as the transported one. This assumption was already implicit in Cabrera and Mott's Theory of oxidation of metals[2] (see also refs. [3–5]), in which the following simple formula was accounting for the activation enthalpy change under electric fields:

$$W^S = \frac{Qd}{2} \cdot E. \tag{1}$$

$W^S$ is the electric work performed by a single ion of charge $Q$ hopping a potential barrier that lies halfway ($d/2$) between its initial and final position ($\frac{Qd}{2}$ is the dipole moment originated from the position change). In this early expression for $W^S$, the electric field, $E$, was, in addition, assumed to be colinear with the jump coordinate. It is worth noting that the displacement, $\frac{d}{2}$, is not a displacement of a whole lattice vector, as in the case of the transported charge.

Equation (1) has been widely adopted since its early formulation, and it is still nowadays applied in many different contexts, or for example, in the very active field of multiscale modeling[6–13], or in the interpretations of experimental data related to oxidation refs. [14–16]. Recent studies have started to question Cabrera–Mott's theory of oxidation of metals in nanostructures[17,18], but not Eq. (1). Even at the level of full first-principle calculations, discussions about the limit of the classical picture are only marginal. For instance, ref. [19] does not report any significant deviation of their results with respect to classical models in the case of MgO.

Materials are, however, composed of many interacting particles. Ionic motion occurs, in principle, in a multidimensional energy landscape in which the migration path up to the barrier might comprise configuration changes. It is, therefore, hard to believe that the electric work is, in general, only dependent on the dipole moment—the position change times the charge—of a single ion. A question naturally arises: is it mere coincidence or the signature of a deeper truth as in the aforementioned case of the oxidation state[1]?

While the classical picture is almost universally adopted in the literature, a few anomalous results have been reported, together with inconclusive attempts to explain them. For instance, ref. [20] revealed anomalous activation energies for the ionic conductivity in tantalum oxide. To fit the experimental data, the charge $Q$ in Eq. (1) was multiplied by an effective parameter. More recently, because of discrepancies between the predicted ionic drift mobility in oxide-based memristors and the measured ones (discrepancies of several orders of magnitude in the ratio between the volatility and the switching time), it was proposed to correct Eq. (1), by including local field effects (the Lorentz field)[21]. Such corrections were, however, strongly criticized as lacking physical ground as commented and documented by Meuffels and Schroeder[22].

Adopting the framework of Modern Theory of Polarization[23–25] and decomposing the macroscopic polarization in terms of contributions from the migrating atom (MA)—the hopping ion—and from the environment atoms, we derive an expression that allows a meaningful representation of the underlying many-particle screening mechanisms. This includes nontrivial contributions from the environment, and provides a definition of the equivalent polarization-work charge, that is, the charge to be multiplied by the MA displacement to obtain the polarization change. We then show that, in general, the equivalent polarization-work charge is not equal to the transported charge, it is not equal to the Born effective charge (BEC) of the MA, and it is not restricted to integer numbers. Our proposed new point of view demonstrates that the success of Eq. (1) can be traced back to subtle system-dependent and symmetry-related compensations between different effects. Environment contributions are crucial for understanding and modeling the polarization-work evolution up to the saddle point, with a corresponding global effect that is critically dependent on the material and the system. Our new point of view is illustrated through two examples: charged oxygen defects in an ionic metal oxide, MgO, and in an iono-covalent semiconductor oxide, SiO$_2$.

## Results

**Equivalent polarization-work charge.** In order to find a rational for the success of Eq. (1), we will exploit BECs and an appropriate polarization decomposition. Details on the derivation are provided in the Supplementary Note 2.

The thermodynamic potential, namely the electric enthalpy, $H$, related to the presence of an external electric field, at fixed atomic positions and up to the first order in the perturbation, is: $H(\vec{E}) = H^{(0)} - \Omega \vec{P} \cdot \vec{E}$, where $H^{(0)}$ is the unperturbed potential, $\vec{P}$ is the macroscopic polarization, and $\Omega$ is the volume of the system.

Along the migration path, the electric enthalpy varies proportionally to the variation of the polarization. This variation is called the polarization work (examples are provided in Supplementary note 1). The work from the initial, 0, to the saddle point, $S$, is therefore:

$$W^S = \int_0^S \Omega \, \vec{d} P \cdot \vec{E} = \Omega \Delta \vec{P}^S \cdot \vec{E}. \tag{2}$$

The BEC of an atom $i$, $Z^*_{i,\alpha\beta}$ is defined as the variation of the system's polarization, $\partial P_\alpha$, with respect to the variation of its position, $\partial r_{i,\beta}$: $Z^*_{i,\alpha\beta} = \frac{\partial P_\alpha}{\partial r_{i,\beta}}$. It is worth noting that the BEC is, in general, a nonsymmetric matrix. As a function of the BECs, the polarization variation up to the saddle point reads:

$$\Delta \vec{P}^S = \int_0^S d\vec{P} = \int_0^S \sum_{i=0}^N \bar{\bar{Z}}^*_i(\lambda) \frac{d\vec{r}_i}{d\lambda} d\lambda, \tag{3}$$

where $\lambda$ is the reaction coordinate, $N$ the total number of atoms, and $d\vec{r}_i$ is the infinitesimal displacement vector of atom $i$.

At this point, the MA and environment contributions can be disentangled by using the BEC of the MA (an alternative decomposition that disentangles ionic and electronic contributions can be found in the Supplementary Note 3):

$$\Delta \vec{P}^S = \int_0^S d\vec{P}_{MA} + \int_0^S d\vec{P}_{env}$$
$$= \int_0^S \bar{\bar{Z}}^*_{MA}(\lambda) \cdot \frac{d\vec{r}_{MA}}{d\lambda} d\lambda + \int_0^S \sum_{(j \neq MA)} \bar{\bar{Z}}^*_j(\lambda) \cdot \frac{d\vec{r}_j}{d\lambda} d\lambda. \tag{4}$$

Equation (4) is still exact. The environment contribution can be efficiently computed as $d\vec{P}_{env} = d\vec{P} - d\vec{P}_{MA}$.

We will now perform a series of approximations to bridge between the Modern Theory of Polarization and the classical picture.

If the migration does not significantly change the chemical nature of the bonding between the species, including the MA, the BECs will remain almost constant along the whole reaction path. The BEC could, therefore, be taken out of the integrals in Eq. (4):

$$\Delta \overrightarrow{P}^S \approx Z^*_{\mathrm{MA}} \cdot \overrightarrow{d}^S_{\mathrm{MA}} + \sum_{(j \neq \mathrm{MA})} Z^*_j \cdot \overrightarrow{d}^S_j, \tag{5}$$

where $\overrightarrow{d}^S_{\mathrm{MA}}$ and $\overrightarrow{d}^S_j$ are, respectively, the integrated displacement vector of the MA and of environment atoms along the path from initial to the saddle point. The assumption of constant BECs along the reaction path is justified for the environment atoms for which chemical environment does not change significantly, while it might be more critical for the MA. If a more accurate polarization work is required, we expect that only a detailed evaluation of the MA BEC should be necessary.

Multiplying and dividing the environment contribution by $\overrightarrow{d}^S_{\mathrm{MA}}$, an expression for an equivalent polarization-work charge matrix can be derived. Component by component it reads

$$Q^*_{\alpha,\beta} = Z^*_{\mathrm{MA},\alpha\beta} + \sum_{(j \neq \mathrm{MA})} Z^*_{j,\alpha\beta} \frac{d^S_{j,\beta}}{d^S_{\mathrm{MA},\beta}}. \tag{6}$$

Because of the perturbation originated by the MA jump, it is more likely that uncompensated dipoles appear along the MA displacement direction. The major contribution to the work is, therefore, expected to come from the polarization component parallel to the MA displacement. Trivially, on the sole MA side, the larger dipole contribution will be along the direction of its displacement (jump), as it is along this direction that the MA moves the most. Similarly, MA neighbors are expected to be dragged or pushed along the same direction as a consequence of the Coulomb interaction. Assuming colinearity between the displacement of the MA and the infinitesimal polarization $\mathrm{d}\overrightarrow{P}$ and assuming that $\mathrm{d}\overrightarrow{P}$ keeps almost the same orientation along the path, up to the saddle point, we obtain an approximate form for the polarization work at the saddle point, that reminds Eq. (1)

$$W^S \approx Q^{\mathrm{eqv}} \cdot |\overrightarrow{d}^S_{\mathrm{MA}}| \cdot |\overrightarrow{E}| \cos \theta, \tag{7}$$

with

$$Q^{\mathrm{eqv}} = \pm \frac{|Q^* \cdot \overrightarrow{d}^S_{\mathrm{MA}}|}{|\overrightarrow{d}^S_{\mathrm{MA}}|}. \tag{8}$$

Because of the colinearity assumption, $\theta$ is now the angle between the MA net displacement $\overrightarrow{d}_{\mathrm{MA}}$ and the field $\overrightarrow{E}$. Equation (8) defines the equivalent polarization-work charge, $Q^{\mathrm{eqv}}$, to be plugged into Eq. (1). The sign is fixed by the sign of the macroscopic polarization.

Equation (6) clearly demonstrates that the variation of the barrier height, $W^S$, and consequently of the drift velocity, under electric fields, is neither directly related to the transported charge nor to the BEC of the MA alone. Furthermore, it is clearly not restricted to integer numbers.

**Charged defect migration in MgO and SiO$_2$.** Let us now see how our sample systems behave with respect to the classical theory, in light of our more complete picture.

For all cases, the projection of the macroscopic polarization variation along the displacement vector of the MA depends quasi-linearly on the MA displacement along the path, as shown in Fig. 1a–c (lower panels). The magnitude of the polarization projection is very close to the modulus of the total polarization, which also exhibits a quasi-linear dependency with $d_{\mathrm{MA}}$. The

perpendicular components are, indeed, very small compared with the parallel one. Therefore, the latter should be very close to be colinear with the MA displacement vector. Details on how the projections are defined can be found in the Supplementary Note 2.

On highly symmetric, compact, and ionic materials, like MgO[19,26] and similar rocksalts, because of the symmetry of the lattice, the movement of environment atoms results in a negligible net displacement from the minimum to the saddle point, $\overrightarrow{d}^S_{\mathrm{env}}$, see Fig. 1a (upper panel) and Fig. 2a. Furthermore, because of the strong ionic nature of the material, Mg and O environment atoms exhibit BECs, see upper panel of Fig. 3a, that are close to their corresponding oxidation level, $+2$ and $-2$, and that results in compensating dipoles along the whole path (the charges are opposite and the net displacements are equal and pointing in the same direction). Therefore, the environment contributions to the total dipole cancel out, and only the displacement and charge of the MA contribute to the polarization work and the variation of the migration barrier, see the bottom panel of Fig. 3a. In addition, because of the cubic symmetry, BEC matrices tend to be symmetric and isotropic with identical eigenvalues, see the upper panel of Fig. 3a. It can therefore be simplified to a scalar. The saddle point lies at half the distance between the initial (0) and final ($F$) configuration, and Eq. (7) simplifies into Eq. (1), with $Q^{\mathrm{eqv}}$ equal to the MA-transported charge, $Q$. Therefore, the classical picture seems to hold because of symmetry-related/material-related compensations. As the vacancy moves in the opposite direction with respect to the MA, the polarization work associated with the defect has opposite sign, leading to $Q \approx +2$.

Net displacements related to environment atoms and BECs are rather more complex in SiO$_2$ than in the simple cubic and ionic crystal, MgO, see Fig. 2b, c. SiO$_2$ polymorphs (excluding high-pressure phases) are three-dimensional assemblies of corner-sharing SiO$_4$ tetrahedra. Oxygen interstitials with charge $-2$ have the structure of two edge-sharing pentahedra (pentacoordination for the two silicon atoms). The migration of the oxygen interstitial involves the hopping of one of the oxygen atoms (MA) that defines the shared edge, see Fig. 1b (upper panel). For the case of the charge $+2$ oxygen vacancy, two separated units can be identified, both made from a back-projected positively charged silicon atom (each unit carries a $+1$ charge) weakly bonded to a three-coordinated back oxygen. The movement of one of these units involves the displacement of the back-projected silicon (MA) that gets attached to a different neighboring oxygen atom, see Fig. 1c (upper panel). Because of the tetrahedral site symmetry and the iono-covalent nature of the bonds, the BEC of oxygen and silicon atoms is strongly anisotropic and its trace does not correspond to the oxidation level of the species.

In the particular case of $\mathrm{I}^{2-}_{\mathrm{O}}$, the BEC matrix is significantly nonisotropic, Fig. 3b (upper panel), with eigenvalue differences of up to $1.5e$, as in the case of networking oxygen atoms. This anisotropy makes it crucial to explicitly account for the tensor nature of the BEC, and its corresponding matrix–vector product with the displacement vector, $Z^* \cdot \overrightarrow{d}^S$. Along the reaction path, the trace of the MA BEC remains almost constant with a signature of charge localization near the saddle point. This stronger localization contributes, however, small variations on the trace of the order of a few tens of $e$. Charge-transfer effects are, therefore, negligible. As a consequence, the MA BEC can be approximated by a constant matrix taking, for instance, the BEC at the initial configuration, $Z^{*,0}$. Figure 2b clearly shows that in the saddle-point neighborhood, the MA displacement is combined with the recoil of mainly silicon environment atoms. The contribution to the polarization coming from the parallel

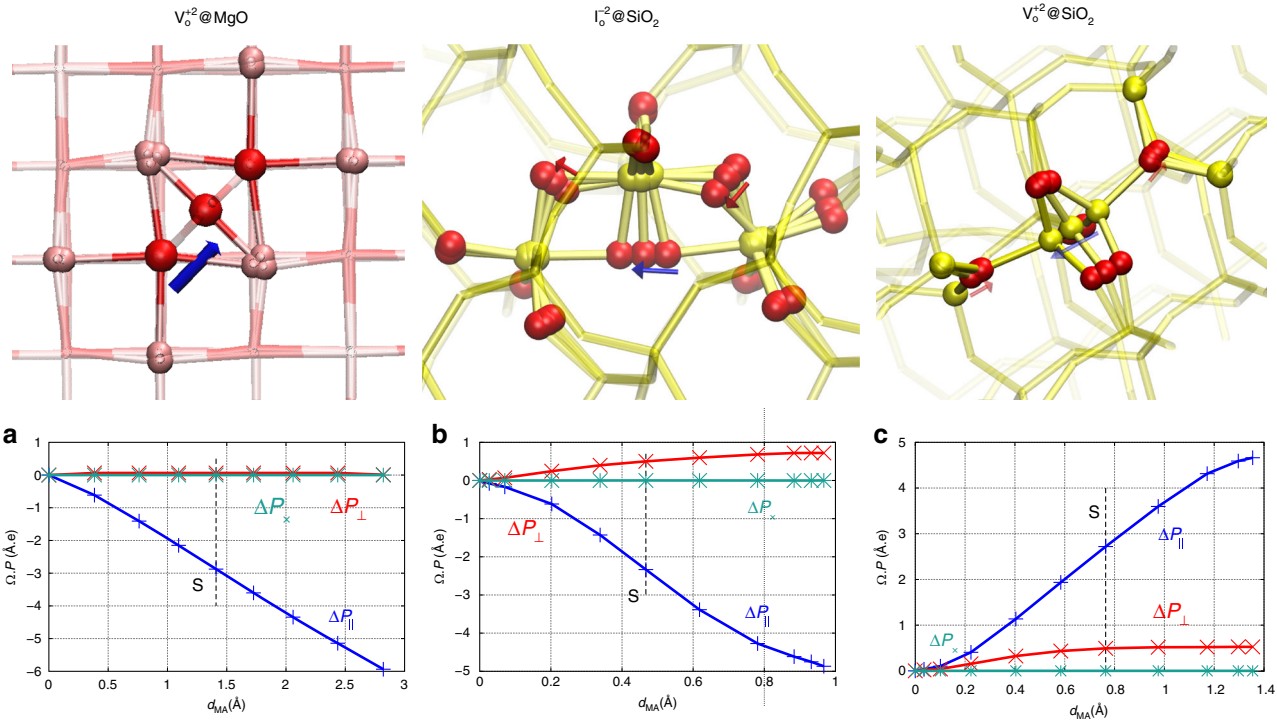

**Fig. 1 Defect migration path and the corresponding macroscopic polarization change from first principles.** The top panels describe the atomic mechanism of the migration of a charged +2 oxygen vacancy, $V_O^{2+}$, in MgO, and of a charged −2 oxygen interstitial, $I_O^{2-}$, and a charged oxygen +2 vacancy, $V_O^{2+}$, in SiO₂. The remaining panels show the parallel ($\Delta P_\parallel$) and perpendicular projections ($\Delta P_\perp$, $\Delta P_\times$) of $\Delta P$ as a function of the net displacement of the migrating atom, $d_{MA}$, along the reaction path, for, respectively, $V_O^{2+}$ in MgO (**a**), $I_O^{2-}$ (**b**), and $V_O^{2+}$ (**c**) in SiO₂. The parallel component corresponds to the projection along the MA displacement vector ($d\,\vec{r}_{MA}^{\,J}$), which depends on the reaction path coordinates at a given point of the path. Further details on how the projections are defined can be found in Supplementary Note 2.

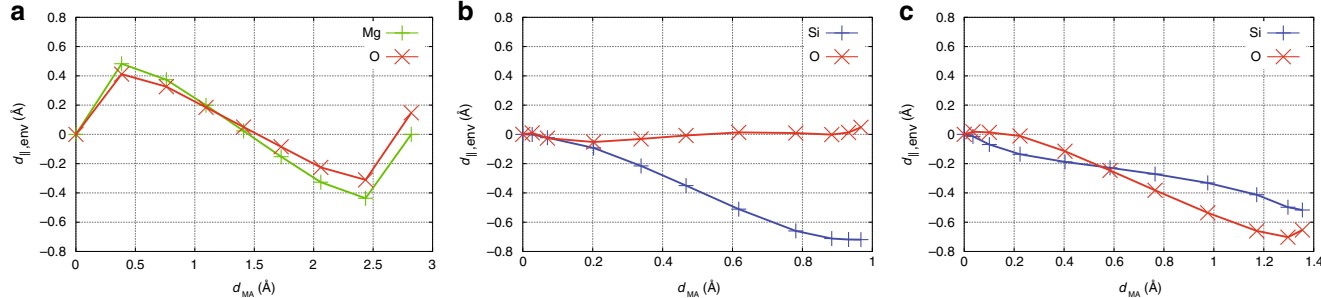

**Fig. 2 Environment atom displacement along defect migration path.** Projection of the total net displacement of environment atoms (for each environment species) along the MA displacement vector as a function of the MA net displacement for, respectively, $V_O^{2+}$ in MgO (**a**), $I_O^{2-}$ (**b**) and $V_O^{2+}$, (**c**) in SiO₂. The projection is computed as $d_{env,\parallel}^{(l)} = \int_0^l |\sum_i^{N-1} \frac{d\vec{r}_i^{(l)}}{d\lambda} d\lambda \frac{d\vec{r}_{MA}^{(l)}}{|d\vec{r}_{MA}^{(l)}|}|$, where $l$ tags a given point along the reaction path, $i$ is the atom index, and $N$ is the total number of atoms.

component of the environment, $\Delta P_{env,\parallel}$ is, indeed, very important and comparable to the contribution of the MA, $\Delta P_{MA,\parallel}$, see Fig. 3b (bottom panel) and Table 1. If only the MA macroscopic polarization component is taken into account, the charge found from the ratio at the saddle point is of about −2.2e, in line with the classical ideas, as there is no charge transfer on the MA. The total macroscopic polarization work is, however, significantly higher, leading to a high negative equivalent polarization-work charge of −5.1e or of 4.9e, depending on whether the whole or the parallel component is considered. Such an anomalously high polarization work is originated from displacements of neighboring atoms, mainly of opposite charge, that is, Si atoms. This results in a significant increase of the polarization that apparently

dresses the MA with a higher equivalent polarization-work charge, similar to a "Quasi-Particle"/"Quasi-ion" picture. In other words, the apparent "dressing" is a many-particle effect that arises from the dielectric response of environment atoms that contributes to the overall macroscopic dipole moment.

Finally, for $V_O^{2+}$, the displacements of oxygen and silicon environment atoms are not trivially compensating each other, see Fig. 2c as in the MgO case. From the minimum to the saddle point, O environment atoms recoil about twice the recoil distance of silicon atoms. However, as the trace of oxygen BECs is close to be half the opposite value of silicon BECs, the contributions to the total dipole of environment atoms cancel out, see also Fig. 3c (bottom panel) and Table 1. As such, only the BEC of the MA and

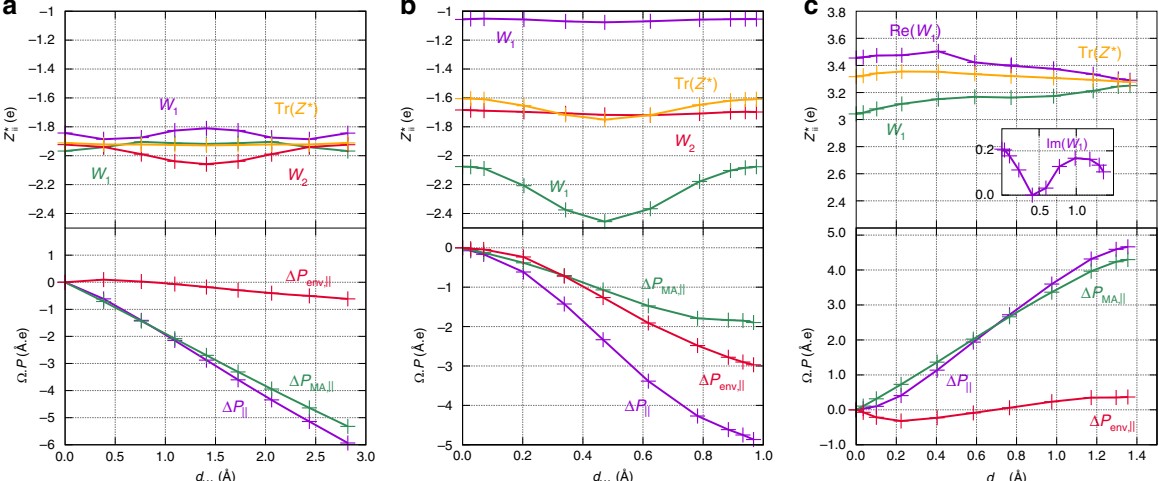

**Fig. 3 First-principles Born effective charges and macroscopic polarization change along defect migration path.** Top panels show the evolution of the eigenvalues and the trace ($Tr(Z^*)$) of the BEC matrix $Z^*_{\alpha\beta}$ of the moving atom (MA) as function of its net displacement ($d_{MA}$), for, respectively, $V_O^{2+}$ in MgO (**a**, the MA is an oxygen atom), $I_O^{2-}$ (**b**, the MA is an oxygen atom), and $V_O^{2+}$ (**c**, the MA is a silicon atom) in $SiO_2$. The inset in **c** shows the imaginary part of eigenvalue $W_1$ (as the BEC is a real matrix but not symmetric, its diagonalization might lead to a real eigenvalue and two complex conjugate eigenvalues; see also ref. [38]). Bottom panels display the evolution of the parallel projection of the total macroscopic polarization, $\Delta P_\parallel$, of the parallel projection of the MA contribution, $\Delta P_{MA,\parallel}$, and of the environment contribution $\Delta P_{env,\parallel}$, as a function of the MA net displacement, $d_{MA}$.

**Table 1 Polarization change computed fully from first-principles (FP) and from different levels of approximations for $V_O^{2+}$ in MgO, and for $I_O^{2-}$ and $V_O^{2+}$ in $SiO_2$.**

| $\Delta P$ | $V_O^{+2}$@MgO | $I_O^{-2}$@SiO$_2$ | $V_O^{+2}$@SiO$_2$ |
|---|---|---|---|
| \|Total\| (FP) | 2.9 | 2.4 | 2.8 |
| MA, $\parallel$ (FP) | −2.7 | −1.1 | 2.7 |
| env, $\parallel$ (FP) | −0.2 | −1.3 | +0.0 |
| Total, $\parallel$ (FP) | −2.9 | −2.3 | 2.7 |
| MA, Eq. (5) | −2.6 | −0.9 | 2.5 |
| env, Eq. (5) | −0.0 | −1.0 | −0.0 |
| Total, Eq. (5) | −2.7 | −1.9 | 2.4 |
| MA, Eq. (5), Tr ($Z^*$)/3 | −2.7 | −0.8 | 2.6 |
| env, Eq. (5), Tr ($Z^*$)/3 | −0.0 | −1.2 | −0.3 |
| Total, Eq. (5), Tr ($Z^*$)/3 | −2.8 | −2.0 | 2.3 |
| Eq. (1), Tr($Z^*_{MA}$)/3 | −2.7 | −0.8 | 2.6 |
| Eq. (1) | −2.8 | −0.9 | 3.1 |

"\|Total\| (FP)" is the modulus of the FP Total polarization change. The labels "MA" and "env" refer to the contribution of the moving atom and the environment, respectively. The symbol "$\parallel$" refers to the polarization component parallel to the MA displacement vector ( $\vec{d}_{MA}$). The specification "Tr($Z^*$)/3" indicates that the isotropic BEC value is used instead of the full BEC matrix. Values for the single label "Eq. (1)" are calculated using the oxidation level of the MA.

its displacement contribute to the polarization work, therefore explaining the agreement between the calculated equivalent polarization-work charge, $3.5e$, and the BEC of the MA.

## Discussion

From the above analysis, we can draw a simple recipe to estimate the polarization work from a zero-field reaction path, short-cutting the burden of a full first-principles Berry Phase calculation over the whole path, at least as a first approximation. The macroscopic polarization change can be estimated assuming isotropic charges and a small path-curvature. Equation (8) then simplifies into a sum of species net displacements (initial and saddle point, only) times their respective oxidation state evaluated from the material isotropic BEC, $1/3Tr(Z^*)$. If environment

atoms net displacement at the saddle point times the oxidation state is close to zero, only the MA contribution is relevant, and Eq. (7) simplifies to the classical Eq. (1). Table 1 summarizes the change in the macroscopic polarization obtained at different levels of approximation. Our three examples show the possible different behaviors: (i) in $V_O^{+2}$@MgO, the classical picture would be adequate since the system is highly symmetric and environmental contribution cancels out, (ii) in $I_O^{-2}$@SiO$_2$, the environment contribution is essential to obtain an accurate description and the classical picture fails completely, (iii) in $V_O^{+2}$@SiO$_2$, the system has low symmetry and, although environment contribution cancels out, the tensor nature of the BEC is required for an accurate description, and a simplified version of Eq. (7) based on isotropic charges gives only semi-quantitative agreement.

In conclusion, we establish a clear distinction between transport charge and equivalent polarization-work charge beyond classical ideas, such as providing a solid ground to improve our understanding and modeling accuracy of electrochemical processes under electric fields. We show that, to understand the variation of the migration barrier under electric fields and the underlying microscopic dipole effects, it is necessary to go beyond the classical idea of a single-moving ion. The knowledge of nominal charges and zero-field reaction barriers alone is in general not sufficient to address/characterize ionic motion under electric fields. From the framework of the Modern Theory of Polarization, we demonstrate that the environment response plays, in general, a crucial role. In cases in which colinearity holds and in which BECs do not vary significantly along the reaction path, it is possible to generalize Eq. (1), still keeping a linear relation between a charge and a displacement, but with a revisited "charge" concept, that is, the equivalent polarization-work charge. The equivalent polarization-work charge is not equal to the transported charge or the BEC of the MA alone, and it is not restricted to integer numbers. The standard BEC charge definition becomes "dressed" by the many-particle environment dipole response, leading to a semiclassical model of screened moving ions. It is worth noting that the "dressing" is not a charge transfer but a pure dipole effect. Owing to the connection that we established between our revisited charge model and the Modern Theory of Polarization, the success of Eq. (1) and similar

simplified classical pictures can be traced back to symmetry-related compensations between environment atoms that can only apply to specific systems. We expect, therefore, strong environment contributions to the ionic conductivity under electric fields in low-symmetry or low-density materials and heterostructures, where environment dipoles compensate each other—for instance, at surfaces and interfaces (cathodes, memristors, and OxRRAM as $SiO_x$), in ceramics and glasses, in bulk phosphates and derivatives (Li-ion battery cathodes as $LiFePO_4$), and in soft-matter solids. In particular, the challenging field of polymer-based electrolytes to be used in future flexible high-performing batteries[27,28] could benefit from our framework in order to better understand the detailed atomic-scale mechanisms underlying the ionic conductivity in these materials. As the calculation of BECs and the polarization work is computationally demanding fully from first principles, in practice, it is possible to proceed step by step by first checking the environment net displacement, the local stoichiometry, and the nature of the bonding, and later by including (or not) each of the approximations that brings from the exact polarization work to our semiclassical approximation.

We hope that the present analysis will spur the inclusion of calculated BEC among the properties reported in popular open-source material database such as NOMAD[29] or Material Clouds[30] and will open the way to a more rational design of materials for electrochemistry.

Our findings will not only impact on theoretical/computational modeling, but also the way experimental measures are analyzed as high drift velocities might be tagged "anomalous" only because of reminding old classical ideas.

## Methods

The calculations presented in this work are all based on Density Functional Theory and Modern Theory of Polarization and the Berry Phase[23,24] as implemented in Quantum ESPRESSO[31,32]. A Perdew–Zunger[33] exchange-correlation functional together with a Norm-conserving pseudopotential from the original QE library has been used. The valence wave functions are expanded into plane waves based on an energy cutoff of 80 Ry (1088 eV)). The Brillouin zone is sampled only at $\Gamma$ k-point (https://www.quantum-espresso.org).

The $SiO_2$, Quartz model is a supercell of 216 atoms with a group symmetry $P3_221$. The model has been fully relaxed with a force and stress convergence threshold of 7 meV/Å. The initial and final defect configurations have been relaxed with the same force convergence threshold.

Reaction paths have been calculated by means of Climbing Image Nudged Elastic Band [34,35] (NEB). The path has been decomposed into nine frames. The convergence threshold for the NEB is 0.1 eV/Å, and the distance between the frames is 0.3 Å, while the mean displacement of the MA between consecutive frames is 0.4 Å.

BECs are computed from the Berry Phase by means of finite differences with a real space grid of 0.1 Å. In the calculation of the polarization, special care has been taken to refer all reported values to the same branch, see refs. [36,37].

## Data availability

The software used for the present study (Quantum ESPRESSO 6.0) is released under GPL and is freely available at https://github.com/QEF/q-e. The pseudopotentials (O.pz-mt. UPF, Si.pz-vbc.UPF, Mg.pz-n-vbc.UPF) are available in the pseudopotential table at www.quantum-espresso.org under the sub-category "original QE pp table." Source data that support the findings of this study, including Quantum ESPRESSO inputs and outputs, and the post-processing script (python notebook) that computes BECs, displacements, and polarization changes, are available as a zip file as Supplementary information. Source data are provided with this paper.

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

## Acknowledgements

We are grateful to Stefano Baroni for useful discussions. High-performance computing resources have been provided by the Arctur company (https://hpc.arctur.si) and by CINECA super computing center (www.cineca.it, grant number ISCRA C HP10CQGOFM). N.S., L.G., and M.V. acknowledge the financial support from the Slovenian Research Agency (research core funding no. P2-0412). N.S., L.M.-S., S.d.G., L.G., and A.H. are active members of the Multiscale and Multi-Model Approach for Materials in Applied Science consortium (MAMMASMIAS consortium), and acknowledge the efforts of the consortium in fostering scientific collaboration.

## Author contributions

P.B. and N.R. proposed the topic. N.S. performed the DFT calculations and related analysis. N.S., L.M-S., and S.d.G. analyzed the results and wrote the paper. A.H., M.V., L.G., and B.S. helped to revise the paper. All authors discussed and commented on the results and on the paper.

## Competing interests

The authors declare no competing interests.
