## [Peer Review File · Nature Communications]

Reviewers' comments:

Reviewer #1 (Remarks to the Author):

The manuscript derives an expression to define the equivalent polarization-work charge and expound the new point by using the two examples of charged oxygen defects in the ionic crystal of MgO and the covalent crystal of SiO₂ by using the Quantum Espresso package. The authors established a clear distinction between transport charge and equivalent polarization-work charge beyond classical ideas, providing a solid ground to improve the understanding and modelling accuracy of electro-chemical processes under electric fields. It is valuable and meaningful for the simulation of the ionic migrations in the metal oxidation, semiconductor and the ionic conduction of solid-state electrolytes, however, there are some questions should be addressed:

1. The authors mentioned the more accuracy of the derived expression compared to the classical models, however, there are no detailed analysis between these two models in the calculated accuracy. The authors should add the comparison analyzations, it is better to give some verification.
2. The authors established a new point for calculation of the ionic migration in the crystal lattice, there are indeed some formula derivation in the supporting information. However, to increase the repeatability and the reliability for the calculation methods, thus, the authors should provide the program of this new method in the supporting documents, so that improving the reproducibility of results for the readers.
3. In the computational details described "The SiO₂ , Quartz model is a supercell of 216 atoms", however, the Fig. 1 just exhibit a portion model of SiO₂. So, I think the authors should provide the full optimized structure of the simulation in the support information. In addition, with respect to the SiO₂, the authors should also provide the description of the construction of MgO and set it in the support information.
4. The symbol of the "Ry" in the description of energy cutoff should transform to the more widely used "eV" for the better understand of readers.
5. In the end of the computational details "In the calculation of the polarization special care has been taken to refer all reported values to the same branch, see Ref. ? ?" the references are missed.

Reviewer #2 (Remarks to the Author):

In this contribution, the authors present a method to determine energy landscape parameters for ion migration in solids (work profiles, etc.). Their approach introduces polarization work contributions from both the migrating ion and its environment, and demonstrates that the representative profile is given by a unique, effective charge parameter. This quantity is notable in that it does not reflect the charge being transported during the migration process, nor does it correspond to the conventional Born Effective Charge for ion rearrangement. Both analytical derivations and numerical support are provided, addressing cases where charged vacancies or interstitials migrate in both symmetric and anisotropic solids. In making these observations, the authors demonstrate that a substantial deviations exist from a reasonably popular Cabrera-Mott model when substantial anisotropies / asymmetries are present.

The derivations in this work are clearly presented and the numerical work demonstrates their assertions. I also appreciate the manner in which the authors reduce a reasonably complex problem to delimited physical process (migration ion vs. environmental effects) and introduce a simple framework. This can be used - alongside limited numerics (depending on the system) - to make meaningful predictions and lend clear physical insight. Last but not least, the qualifications imposed on the widely

used Cabrera-Mott theory are very important and should be impactful.

I would recommend this manuscript pending a few points of clarification:

- 1) The authors qualify dynamics when charge localization occurs at a saddle point for migration, leading to small variations in the Born Effective Charge. This allows the BEC to be approximated by its initial value. Nonetheless, strong localization is far from universal in anisotropic solids. It would be useful if the authors could formally qualify the utility of this approximation in a more general case (i.e., how much variation can be tolerated for this initial value assumption to be applicable). An answer to this question would help qualify which simplifying approximations can be made from the outset given only preliminary calculations. Furthermore, this would support how often expensive calculations can be avoided, as the authors claim.
- 2) The scope of applications is only weakly stated, and only sparse connections are drawn to the literature (with respect to solid state electrolytes, metal / semiconductor oxidation etc). This should not be exhaustive, however, the authors' presentation more closely resembles the discussion in a specialist journal.
- 3) The caption in Fig. 2 does not reflect the labels [(a), (b), (c)] stated above the figure. The precise physical system and calculation being presented should be stated clearly in this and all other figure captions.
- 4) A similar issue exists with Fig. 3, in that it is difficult to interpret the figure from the caption. Here, the meaning / content of the figure should be stated first, followed by a discussion of what each component represents, followed by definitions etc. This is a global problem - the figures are difficult to interpret without reading the referencing segment of the manuscript.
- 5) There are missing references at the end of the methods.

We thank the two Referees for their very positive evaluation of our work. A number of remarks have been raised which we have addressed in the revised version we are submitting. We detail in the following our answers to the Referees' remarks and the action taken.

For clarity Referees' remarks are reported in red followed by our reply.

Reviewer #1 (Remarks to the Author):

The manuscript derives an expression to define the equivalent polarization-work charge and expound the new point by using the two examples of charged oxygen defects in the ionic crystal of MgO and the covalent crystal of SiO₂ by using the Quantum Espresso package. The authors established a clear distinction between transport charge and equivalent polarization-work charge beyond classical ideas, providing a solid ground to improve the understanding and modelling accuracy of electro-chemical processes under electric fields. It is valuable and meaningful for the simulation of the ionic migrations in the metal oxidation, semiconductor and the ionic conduction of solid-state electrolytes, however, there are some questions should be addressed:

We thank the Referee for His/Her positive appraisal. See the following for the specific points.

1. The authors mentioned the more accuracy of the derived expression compared to the classical models, however, there are no detailed analysis between these two models in the calculated accuracy. The authors should add the comparison analyzations, it is better to give some verification.

As suggested by the Referee we added to the manuscript a detailed comparison of the classical model and our newly derived expression. In order to make the comparison clearer we included Table 1 in the discussion section with numerical comparison between different approximations.

2. The authors established a new point for calculation of the ionic migration in the crystal lattice, there are indeed some formula derivation in the supporting information. However, to increase the repeatability and the reliability for the calculation methods, thus, the authors should provide the program of this new method in the supporting documents, so that improving the reproducibility of results for the readers.

We thank the referee for pointing out the reproducibility issue. We use a script for computing the MA and environment displacements. We included a zip file as supporting information that contains a new version of the script (more user friendly, with a basic documentation inside) and all the raw inputs and outputs (Quantum ESPRESSO format) required to reproduce and validate our calculations.

3. In the computational details described "The SiO₂ , Quartz model is a supercell of 216 atoms", however, the Fig. 1 just exhibits a portion model of SiO₂. So, I think the authors should provide

the full optimized structure of the simulation in the support information. In addition, with respect to the SiO₂, the authors should also provide the description of the construction of MgO and set it in the support information.

The raw inputs and outputs included in the zip file contain the whole structural models (see previous referee comment)

4. The symbol of the “Ry” in the description of energy cutoff should transform to the more widely used “eV” for the better understanding of readers.

Ry are the energy units used by the Quantum ESPRESSO code used in our work and is the natural unit for its user basis. In order to help readers more familiar with other codes we also provide as the Referee suggests the equivalent value in eV .

5. At the end of the computational details “In the calculation of the polarization special care has been taken to refer all reported values to the same branch, see Ref. ? ?” the references are missed.

We thank the Referee for noticing the problem. Now the references appear properly.

Reviewer #2 (Remarks to the Author):

In this contribution, the authors present a method to determine energy landscape parameters for ion migration in solids (work profiles, etc.). Their approach introduces polarization work contributions from both the migrating ion and its environment, and demonstrates that the representative profile is given by a unique, effective charge parameter. This quantity is notable in that it does not reflect the charge being transported during the migration process, nor does it correspond to the conventional Born Effective Charge for ion rearrangement. Both analytical derivations and numerical support are provided, addressing cases where charged vacancies or interstitials migrate in both symmetric and anisotropic solids. In making these observations, the authors demonstrate that a substantial deviation exists from a reasonably popular Cabrera-Mott model when substantial anisotropies / asymmetries are present.

The derivations in this work are clearly presented and the numerical work demonstrates their assertions. I also appreciate the manner in which the authors reduce a reasonably complex problem to delimited physical process (migration ion vs. environmental effects) and introduce a simple framework. This can be used - alongside limited numerics (depending on the system) - to make meaningful predictions and lend clear physical insight. Last but not least, the qualifications imposed on the widely used Cabrera-Mott theory are very important and should be impactful.

We thank the Referee for His/Her positive evaluation of our work.

I would recommend this manuscript pending a few points of clarification:

1) The authors qualify dynamics when charge localization occurs at a saddle point for migration, leading to small variations in the Born Effective Charge. This allows the BEC to be approximated by its initial value. Nonetheless, strong localization is far from universal in anisotropic solids. It would be useful if the authors could formally qualify the utility of this approximation in a more general case (i.e., how much variation can be tolerated for this initial value assumption be applicable). An answer to this question would help qualify which simplifying approximations can be made from the outset given only preliminary calculations. Furthermore, this would support how often expensive calculations can be avoided, as the authors claim.

In order to address Referee's concerns we added in the manuscript, after Eq 5, a comment discussing the assumption of constant BECs. We do expect, as stated now in the manuscript, that environment atoms BECs do not vary significantly along the reaction path as their chemical environment is only slightly modified. The change of MA BEC might be more critical, but the choice can be improved, if necessary, by interpolating its value at initial and saddle point. In general, change in charge localization or bond rehybridization will be responsible for the variation of the MA BEC at the saddle point. In cases in which there is no rehybridization (as for the MgO case), the BEC will remain almost unaffected and the approximation of taking the initial BEC value is fully justified.

We added Table 1 in the discussion section that summarizes the polarization change for the three processes examined according to the different approximations discussed in the manuscript. These three examples show the different possibilities: i) in V@MgO the classical picture would be adequate since the system is highly symmetric and environmental contribution cancels out, ii) in I_O@SiO2 the environment contribution is essential to obtain an accurate description and the classical picture fails completely, iii) in V@SiO2 the system has low symmetry and although environment contribution cancels out the tensorial nature of the BEC is required for an accurate description. We changed the manuscript accordingly.

We would like to stress that the main focus of the present manuscript is to provide a sound theoretical framework for the polarization work in terms of the Modern Theory of Polarization. When and where this can be reduced to a simplified model is something that has to be assessed case by case, as discussed in the discussion and conclusion sections. Our framework provides the tools to detect and correct failures of the classical description.

2) The scope of applications is only weakly stated, and only sparse connections are drawn to the literature (with respect to solid state electrolytes, metal / semiconductor oxidation etc). This should not be exhaustive, however, the authors' presentation more closely resembles the discussion in a specialist journal.

Following Referee's suggestion in the conclusions/perspectives section of the manuscript we have added a paragraph listing the many systems in which a reexamination of ionic conductivity

on the basis of our new theoretical framework may be important. We further suggest that the inclusion of BEC reference values in material entries in open databases (such as NOMAD or Material clouds) may lead to advance in the computer assisted design of materials for solid electrochemistry.

We also expanded a bit the introduction including previous attempts to rationalize “anomalous” ionic conductivity.

3) The caption in Fig. 2 does not reflect the labels [(a), (b), (c)] stated above the figure. The precise physical system and calculation being presented should be stated clearly in this and all other figure captions.

4) A similar issue exists with Fig. 3, in that it is difficult to interpret the figure from the caption. Here, the meaning / content of the figure should be stated first, followed by a discussion of what each component represents, followed by definitions etc. This is a global problem - the figures are difficult to interpret without reading the referencing segment of the manuscript.

We carefully revised the captions of the figures and fixed them according to Referee’s suggestion.

5) There are missing references at the end of the methods.

We thank the Referee for noticing the problem. Now the references appear properly.

REVIEWERS' COMMENTS:

Reviewer #2 (Remarks to the Author):

The authors' response has satisfied my scientific concerns / comments. It should be accepted for publication pending minor revisions:

1. The y-axis label should appear next to the axis for each panel in the figures, unless they are placed directly side-by-side (with no interstitial space between panels).
2. The axis ranges for the inset of Fig. 2c are not apparent (particularly for the x-axis).
3. Angs should not be used for units in the axis labels. The angstrom symbol is preferred (\AA in latex).

A few other minor typos / errors that I caught (most of which were introduced during revisions). Most of these would likely be resolved at the production stage, but it might help for clarity to resolve them now.

Parallel should not be capitalized in the caption of Fig. 1.

Last paragraph, page 1.

"un-conclusive attempts to explain them. For instance, Ref.20 reviled"

should read

"inconclusive attempts to explain them. For instance, Ref.20 revealed"

Last paragraph, page 2.

" is more justified for the environment atoms for which chemical environment does not change significantly, "

should read

" is justified for environment atoms where the chemical environment does not change significantly"

Top paragraph, second column of page 5.

"symmetry and although environment contribution cancels out the tensor nature of the BEC is required for an accurate description "

should read

"symmetry and, although the environment contribution cancels out, the tensor nature of the BEC is required for an accurate description "

page 6:

"We, expect, therefore, strong environment contributions to the ionic conductivity under electric fields in low symmetry or low density materials and heterostructures, where environment dipoles can not compensate each others, like,for instance, at surfaces and interfaces"

should read:

"We expect, therefore, strong environment contributions to the ionic conductivity under electric fields in low symmetry or low density materials and heterostructures, where environment dipoles can not compensate each other --- for instance, at surfaces and interfaces"